# RTLIO: Real-Time LiDAR-Inertial Odometry and Mapping for UAVs

**DOI:** 10.3390/s21123955

**Published:** 2021-06-08

**Authors:** Jung-Cheng Yang, Chun-Jung Lin, Bing-Yuan You, Yin-Long Yan, Teng-Hu Cheng

**Affiliations:** Department of Mechanical Engineering, National Yang Ming Chiao Tung University, Hsinchu 30010, Taiwan; johnsongash.gdr07g@nctu.edu.tw (J.-C.Y.); chadlin.gdr07g@nctu.edu.tw (C.-J.L.); physical31031.c@nycu.edu.tw (B.-Y.Y.); yanlong658.gdr08g@nctu.edu.tw (Y.-L.Y.)

**Keywords:** LiDAR-inertial odometry, state estimation, sensor fusion, SLAM

## Abstract

Most UAVs rely on GPS for localization in an outdoor environment. However, in GPS-denied environment, other sources of localization are required for UAVs to conduct feedback control and navigation. LiDAR has been used for indoor localization, but the sampling rate is usually too low for feedback control of UAVs. To compensate this drawback, IMU sensors are usually fused to generate high-frequency odometry, with only few extra computation resources. To achieve this goal, a real-time LiDAR inertial odometer system (RTLIO) is developed in this work to generate high-precision and high-frequency odometry for the feedback control of UAVs in an indoor environment, and this is achieved by solving cost functions that consist of the LiDAR and IMU residuals. Compared to the traditional LIO approach, the initialization process of the developed RTLIO can be achieved, even when the device is stationary. To further reduce the accumulated pose errors, loop closure and pose-graph optimization are also developed in RTLIO. To demonstrate the efficacy of the developed RTLIO, experiments with long-range trajectory are conducted, and the results indicate that the RTLIO can outperform LIO with a smaller drift. Experiments with odometry benchmark dataset (i.e., KITTI) are also conducted to compare the performance with other methods, and the results show that the RTLIO can outperform ALOAM and LOAM in terms of exhibiting a smaller time delay and greater position accuracy.

## 1. Introduction

### 1.1. Background

Precise ego-motion estimation and active perception play important roles when performing navigation tasks or exploring unknown environments in robotics applications, and the potential of small unmanned airborne (S-UAS) platforms applied to collect remote sensing data have been analyzed [1]. Unmanned aerial vehicles (UAVs) running simultaneous localization and mapping (SLAM) algorithms can also be used to perform numerous tasks, including surveillance, rescue, and transportation in extreme environments [2,3,4]. In the field of SLAM, the performance of state estimation is highly reliant on sensors, such as cameras, LiDAR, and inertial measurement units (IMUs). However, there are limitations associated with each type of sensor, such as minimum illumination requirements and the presence of noise. To overcome these shortcomings of stand-alone sensors, multiple sensors have been used to increase the reliability of estimation [5,6,7,8,9]. The methods utilizing multiple sensors for state estimation are categorized into two types: loosely coupled (cf. [5,6]) and tightly coupled (cf. [7,8,9]). The tightly coupled approach directly fuses LiDAR and inertial measurements through a joint optimization that minimizes some residuals, whereas the loosely coupled approach deals with the multiple sensors separately. The tightly coupled method is less computationally efficient and more difficult to implement than the loosely coupled approach, but it is more robust in its approach to noise and more accurate [8].

Accurate and real-time localization is crucial to the feedback control of UAVs in practical applications. Acquiring accurate localization information by solving the tightly coupled problem requires a considerable amount of computation, which decreases the frequency at which state estimation can be performed for providing real-time feedback. Moreover, the requirements of robust, precise, and fast localization increase the difficulty of designing algorithms. The visual inertial odometry (VIO) method [7] proposes achieving precise and real-time results based on tightly coupled VIO that fuses camera and IMU measurements for state estimation. However, its performance can be impaired by poor lighting conditions. Since 3D LiDAR sensors are less influenced by lighting conditions and can also provide range measurements of the surrounding environment, they have been successfully used for ego-motion estimation [8,9,10,11]. Most LiDAR systems update at a lower frequency than cameras (usually 10 Hz), which means that the point cloud can be distorted when the LiDAR moves aggressively. In contrast to LiDAR, an IMU is capable of extremely high update rates, and so combining LiDAR and IMU allows their individual deficiencies to be compensated, and the state estimation for UAVs can be solved by tightly coupled optimization.

In [11], the feature points were extracted from the LiDAR point cloud, and the corresponding features from the last LiDAR measurement were matched to estimate the ego-motion. To refine the odometry, feature points were matched and registered to the feature maps. In [9], it was shown that using tightly coupled LiDAR inertial odometry (LIO) with multiple window frames to local map is too time-consuming, while the accuracy is significantly degraded if there are too few windows. The approach taken in [7] cannot be used, either in a dark or highly dynamic illumination environment. Therefore, there is always a trade-off between high accuracy and computation efficiency, and the localization performance using other sensors (e.g., cameras) to reduce the computation loading can be affected by environmental factors.

### 1.2. Related Works

In [11], IMUs play an important role by providing an initial guess before performing estimation. However, this approach mainly relies on LiDAR information to estimate the motion, by matching feature points extracted from the local surface to the corresponding feature, estimating the relative transform, and constructing a global map. In [7], the gyroscope bias, scale, and direction of gravity can be corrected through the initialization step. In [12], the estimator combines IMU data and plane features obtained from LiDAR for joint optimization. Notably, feature planes are compressed into the closest point in each frame, so that the estimator is able to run in real time. In [9], tightly coupled 3D LIO for graph optimization was demonstrated in both indoor and outdoor environments, but the estimation process still took too long. In [13], a feature extraction algorithm for LiDAR systems was proposed with small fields of view, and feature points were used to estimate odometry and mapping. Moreover, linear interpolation was used to suppress the effect of motion blur associated with LiDAR movement, with each point in the same frame being compensated for this movement.

Loop closure is an approach that corrects the drift that occurs during long-term operation. The method starts by identifying previously visited places, and the iterative closest point (ICP) is the most common approach that involves searching the matches between the current laser scan and the existing map. Another method computes feature descriptors from laser scans, and then verifies loop closure based on certain conditions. In [10], features were segmented into many clusters, which enables the method to perform real-time pose estimation, even in a large-scale environment. In the back-end, the *k*-dimensional (KD)-tree method was used to search for the closest keyframe when performing loop closure, with loop closure established once the residual from ICP is sufficiently small. In [14], a real-time mapping approach was proposed that involved inserting laser scans into a probability grid. In that method, a branch-and-bound search runs in the back-end for loop closure, and if a sufficient match is found in the search window, the loop closure constraint is added to the optimization problem. Overall, this system achieves a moderate capability and pixel-level accuracy using 2D LiDAR, but it is still too time-consuming for applying to 3D LiDAR data. The approach in [15] classified laser scans into segments with feature descriptors, and the transformation was obtained by matching these segments with the map. This method saves time compared to matching the entire laser scan, but its performance may be highly dependent on the accuracy of the classifier.

A tightly coupled LIO is developed in this work to obtain high-accuracy and high-frequency localization output for the feedback control of UAVs. Although tightly coupled methods normally require more computation loading, the developed approach can generate more-accurate and more-frequent localization information. The frame-to-map estimation process is robust and stable, and loop closure is applied to further correct the accumulated error when a loop is detected. Here, the performance of this new method is compared with other approaches in the literature using the publicly available KITTI (http://www.cvlibs.net/datasets/kitti/eval_odometry.php, accesed on 5 April 2021) dataset [16,17]. KITTI dataset was chosen, since it is the first dataset that provides accurate ground truth. The data were collected using a Velodyne HDL-64E laser scanner that produces more than one million 3D points per second and a state-of-the-art OXTS RT 3003 localization system which combines GPS, GLONASS, an IMU and RTK correction signals. Additionally, a fair comparison is possible using KITTI dataset due to its large scale nature as well as the proposed novel metrics, which capture different sources of error by evaluating error statistics over all sub-sequences of a given trajectory length or driving speed [16]. Many works (e.g., [10,11]) also used KITTI as benchmarks to evaluate the localization accuracy. The results demonstrate that the algorithm applied in this work can outperform other approaches in terms of both accuracy and frequency. The main contributions of this study are summarized as follows:IMU excitation is not required for initialization, in contrast to [7].Online relocalization combined with loop closure and pose-graph optimization methods have been developed for odometry and mapping that are more accurate than in [9].In contrast to the odometry and mapping algorithm [11], the developed RTLIO can provide a high-frequency of odometry for the UAV and constructing maps synchronously.

### 1.3. Overview

The architecture of RTLIO is shown in Figure 1. The system starts with measurement preprocessing (Sect. IV), in which point clouds from the LiDAR measurement are classified into corner points and plane points. The distorted clouds are corrected by the integration of IMU measurements between two consecutive LiDAR frames. In the front-end (Sect. V, VI), the initialization processing provides the bias of the gyroscope, direction of gravity, and initial velocity for bootstrapping the subsequent nonlinear optimization-based RTLIO. In the sliding window optimization, the cost function is constructed to include the marginalization, LiDAR, and IMUs information for solving the UAV pose. In the back-end (Sect. VII), the loop closure is used to detect whether the current position has been revisited, and the pose graph optimization module is used to reduce the accumulated drift to increase positioning accuracy. Finally, RTLIO provides two frequency poses. One is the LiDAR-rate pose after preprocessing and optimization at 10 Hz; the other is the IMU-rate pose generated by the IMU propagation in RTLIO at 400 Hz.

Let body frame $b_{k}$ be defined on the IMU, where *k* denotes the frame when the $k^{\mathrm{th}}$ LiDAR measurement is acquired. The world frame *w* is defined on the initial body frame, and the direction of the gravity is aligned with the z axis of the world frame. The LiDAR frame *l* be defined on the LiDAR. The rotation from frame *A* to frame *B* is denoted as $q_{A}^{B}$ or $R_{A}^{B},$ and the translation transforming from frame *A* to frame *B* is denoted, as $p_{B}^{A}$. ⊗ represents the Hamilton product between two quaternions. All other variables are listed in Table 1.

## 2. Methodology

### 2.1. Measurement Preprocessing

#### 2.1.1. Time Alignment

The time stamps of the measurements from the LiDAR and camera are illustrated in Figure 2, where the sliding window includes the latest *m* LiDAR frames and each frame contains a set of IMU measurements, since the sensing rate of the IMU is much higher (e.g., 200 Hz).

#### 2.1.2. IMU Preintegration

IMU preintegration and the covariance matrix derivation with the continuous-time IMU dynamics of an error-state Kalman filter were proposed in [7,9,18,19]. Based on [20], the IMU states can be divided into true states *X*, nominal states $\hat{X},$ and error states δX, whose compositions are defined as
(1)$$\begin{array}{cc} X & =\hat{X}⊞\delta X\\ X & ={\left[\begin{array}{ccccc} \alpha , & \theta , & \beta , & b_{a}, & b_{\omega } \end{array}\right]}^{T}, \end{array}$$
where α, θ, and β are the position, orientation, and velocity, respectively, and $b_{a}$ and $b_{\omega }$ are defined in Table 1. The operation ⊞ for a state *v* in the vector space is simply the Euclidean addition, i.e., $v=\hat{v}+\delta v,$ and for quaternion, it implies the multiplication of the quaternions, i.e., ⊗.

Measurements ${\hat{a}}_{t}$ and ${\hat{\omega }}_{t}$ at time $t\in \left(t_{k},\;t_{k+1}\right]$ are defined as
(2)$$\begin{array}{cc} {\hat{a}}_{t} & =a_{t}+b_{a_{t}}+R_{w}^{t}g^{w}+n_{a}\\ {\hat{\omega }}_{t} & =\omega_{t}+b_{\omega_{t}}+n_{\omega }\\ n_{a} & ∼N(0,\;\sigma_{a}^{2})\\ n_{\omega } & ∼N(0,\;\sigma_{\omega }^{2}), \end{array}$$
where $n_{a}$ and $n_{\omega }$ are defined as random variables with normal distribution (i.e, N) with zero mean and variances $\sigma_{a}^{2}$ and $\sigma_{\omega }^{2}$.

The position, velocity, and orientation states between two body frames $b_{k}$ and $b_{k+1}$ can be propagated by integrating IMU measurements ${\hat{a}}_{t}$ and ${\hat{\omega }}_{t}$ during $t\in \left(t_{k},\;t_{k+1}\right]$ in the world frame as
(3)$$\begin{array}{cc} p_{b_{k+1}}^{w} & =p_{b_{k}}^{w}+v_{b_{k}}^{w}\Delta t_{k}\\ \; & +\int \int_{t\in [t_{k},\;t_{k+1}]}\left(R_{t}^{w}\left({\hat{a}}_{t}-b_{a_{t}}-n_{a}\right)-g^{w}\right)dt^{2}\\ v_{b_{k+1}}^{w} & =v_{b_{k}}^{w}+\int_{t\in [t_{k},\;t_{k+1}]}\left(R_{t}^{w}\left({\hat{a}}_{t}-b_{a_{t}}-n_{a}\right)-g^{w}\right)dt\\ q_{b_{k+1}}^{w} & =q_{b_{k}}^{w}⊗\int_{t\in [t_{k},\;t_{k+1}]}\left[\begin{array}{c} 0\\ \frac{1}{2}\Omega \left({\hat{\omega }}_{t}-b_{\omega_{t}}-n_{\omega }\right) \end{array}\right]\gamma_{t}^{bk}dt, \end{array}$$
where Ω is the same as defined in (3) of [7]. Transforming (Equation 3) from the world frame to frame $b_{k}$ yields
(4)$$\begin{array}{cc} R_{w}^{b_{k}}p_{b_{k+1}}^{w} & =R_{w}^{b_{k}}\left(p_{b_{k}}^{w}+v_{b_{k}}^{w}\Delta t_{k}-\frac{1}{2}g^{w}\Delta t_{k}^{2}\right)+\alpha_{b_{k+1}}^{b_{k}}\\ R_{w}^{b_{k}}v_{b_{k+1}}^{w} & =R_{w}^{b_{k}}\left(v_{b_{k}}^{w}-g^{w}\Delta t_{k}\right)+\beta_{b_{k+1}}^{b_{k}}\\ \gamma_{b_{k+1}}^{b_{k}} & ={\left[q_{b_{k}}^{w}\right]}^{-1}⊗q_{b_{k+1}}^{w}, \end{array}$$
where $\alpha_{b_{k+1}}^{b_{k}}$, $\beta_{b_{k+1}}^{b_{k}}$, and  $\gamma_{b_{k+1}}^{b_{k}}$ are the true states of the IMU integration and $\gamma_{b_{k+1}}^{b_{k}}$ is the quaternion form of $\theta_{b_{k+1}}^{b_{k}}$ defined as
(5)$$\begin{array}{cc} \alpha_{b_{k+1}}^{b_{k}} & =\int \int_{t\in [t_{k},\;t_{k+1}]}R\left(\gamma_{t}^{b_{k}}\right)\left({\hat{a}}_{t}-b_{a_{t}}-n_{a}\right)dt^{2}\\ \beta_{b_{k+1}}^{b_{k}} & =\int_{t\in [t_{k},\;t_{k+1}]}R\left(\gamma_{t}^{b_{k}}\right)\left({\hat{a}}_{t}-b_{a_{t}}-n_{a}\right)dt\\ \gamma_{b_{k+1}}^{b_{k}} & =\int_{t\in [t_{k},\;t_{k+1}]}{\left[\begin{array}{c} 0\\ \frac{1}{2}\left({\hat{\omega }}_{t}-b_{\omega_{t}}-n_{\omega }\right) \end{array}\right]}_{R}\gamma_{t}^{b_{k}}dt. \end{array}$$

The noises in (Equation 5) are unknown, and so the nominal states can be expressed as
(6)$$\begin{array}{cc} {\hat{\alpha }}_{b_{k+1}}^{b_{k}} & =\int \int_{t\in [t_{k},\;t_{k+1}]}R\left({\hat{\gamma }}_{t}^{b_{k}}\right)\left({\hat{a}}_{t}-{\hat{b}}_{a}\right)dt^{2}\\ {\hat{\beta }}_{b_{k+1}}^{b_{k}} & =\int_{t\in [t_{k},\;t_{k+1}]}R\left({\hat{\gamma }}_{t}^{b_{k}}\right)\left({\hat{a}}_{t}-{\hat{b}}_{a}\right)dt\\ {\hat{\gamma }}_{b_{k+1}}^{b_{k}} & =\int_{t\in [t_{k},\;t_{k+1}]}{\left[\begin{array}{c} 0\\ \frac{1}{2}\left({\hat{\omega }}_{t}-{\hat{b}}_{\omega }\right) \end{array}\right]}_{R}{\hat{\gamma }}_{t}^{bk}dt, \end{array}$$
where ${\hat{b}}_{a}$ and ${\hat{b}}_{\omega }$ are the biases in the accelerometer and gyroscope.

The difference between the nominal states and the true states is minimized by correcting the nominal states, as described in Section 2.1.3.

#### 2.1.3. Correction of Preintegration

Based on (Equation 1), the error state can be rewritten as
(7)$$\delta X=X⊟\hat{X},$$
where the operation ⊟ for a state *v* in the vector space is simply the Euclidean addition, i.e., $v=\hat{v}-\delta v,$ and for quaternion, it implies the multiplication of the inverse of the quaternion.

Taking the time derivatives of (Equation 5)–(Equation 7) yields
(8)$$\begin{array}{cc} {\dot{\delta X}}_{i} & =F_{i}\delta X_{i}+V_{i}n\\ \left[\begin{array}{c} {\dot{\delta \alpha }}_{i}\\ {\dot{\delta \theta }}_{i}\\ {\dot{\delta \beta }}_{i}\\ {\dot{\delta b}}_{a_{i}}\\ {\dot{\delta b}}_{\omega_{i}} \end{array}\right] & =F_{i}\left[\begin{array}{c} \delta \alpha_{i}\\ \delta \theta_{i}\\ \delta \beta_{i}\\ \delta b_{a_{i}}\\ \delta b_{\omega_{i}} \end{array}\right]+V_{i}\left[\begin{array}{c} n_{a0}\\ n_{\omega 0}\\ n_{a1}\\ n_{\omega 1}\\ n_{b_{a}}\\ n_{b_{\omega }} \end{array}\right]\\ n_{b_{a}} & ∼N(0,\;\sigma_{b_{a}}^{2})\\ n_{b_{\omega }} & ∼N(0,\;\sigma_{b_{\omega }}^{2}), \end{array}$$
where $F_{i}$ and $V_{i}$ are error state dynamics matrices, $n_{a0}$ and $n_{a1}$ are the acceleration noises, $n_{\omega 0}$ and $n_{\omega 1}$ are the angular velocity noises, $n_{b_{a}}$ and $n_{b_{\omega }}$ are modeled as random walks applied to the biases, and $\sigma_{b_{a}}^{2}$ and $\sigma_{b_{\omega }}^{2}$ are variances of $n_{b_{a}}$ and $n_{b_{\omega }},$ respectively.

Based on (Equation 8), the relation between error states $\delta X_{i}$ and $\delta X_{i+1}$ can be discretized as
(9)$$\begin{array}{cc} \delta X_{i+1} & =\delta X_{i}+{\dot{\delta X}}_{i}\delta t\\ \; & =\delta X_{i}+\left(F_{i}\delta X_{i}+V_{i}n\right)\delta t\\ \; & =\left(I+F_{i}\delta t\right)\delta X_{i}+\left(V_{i}\delta t\right)n, \end{array}$$
which describes the relation of two error states at $t_{i}$ and $t_{i+1}$, which can be extended to the two error states at $t_{k+1}$ and $t_{k}$ by
(10)$$\begin{array}{cc} \delta X_{b_{k+1}} & =F^{\prime }\delta X_{b_{k}}+V^{\prime }n\\ F^{\prime } & =\prod_{i=|B_{k+1}|-1}^{1}\left(I+F_{i}\delta t\right)\\ V^{\prime } & =(V_{N-1}\delta t)n\\ \; & +(\sum_{i=|B_{k+1}|-2}^{1}\left(\prod_{j=|B_{k+1}|-1}^{i+1}\left(I+F_{j}\delta t\right)\right)V_{i}\delta t)n. \end{array}$$

According to [18], covariance matrix $Q_{b_{k+1}}^{b_{k}}$ of $\delta X_{b_{k+1}}$ can be computed recursively using the first-order discrete-time covariance updated with the initial value $Q_{b_{k}}^{b_{k}}=0$:(11)$$Q_{i+1}^{b_{k}}=\left(I+F_{i}\delta t\right)Q_{i}^{b_{k}}\left(I+F_{i}\delta t\right){}^{T}+\left(V_{i}\delta t\right)O{\left(V_{i}\delta t\right)}^{T},$$
where *O* contains the diagonal covariance matrices $\sigma_{a_{0}}^{2},\;\sigma_{\omega_{0}}^{2},\;\sigma_{a_{1}}^{2},\;\sigma_{\omega_{1}}^{2},\;\sigma_{b_{a}}^{2},$ and $\sigma_{b_{\omega }}^{2}$. Based on (Equation 1), (Equation 6), and (Equation 10), the corrected preintegrations denoted as ${\bar{\alpha }}_{b_{k+1}}^{b_{k}},$${\bar{\beta }}_{b_{k+1}}^{b_{k}},$ and ${\bar{\gamma }}_{b_{k+1}}^{b_{k}}$ are defined as
(12)$$\begin{array}{cc} {\bar{\alpha }}_{b_{k+1}}^{b_{k}} & ={\hat{\alpha }}_{b_{k+1}}^{b_{k}}+J_{b_{a}}^{\alpha }\delta b_{a_{k}}+J_{b_{\omega }}^{\alpha }\delta b_{\omega_{k}}\\ {\bar{\beta }}_{b_{k+1}}^{b_{k}} & ={\hat{\beta }}_{b_{k+1}}^{b_{k}}+J_{b_{a}}^{\beta }\delta b_{a_{k}}+J_{b_{\omega }}^{\beta }\delta b_{\omega_{k}}\\ {\bar{\gamma }}_{b_{k+1}}^{b_{k}} & ={\hat{\gamma }}_{b_{k+1}}^{b_{k}}⊗\left[\begin{array}{c} 1\\ \frac{1}{2}J_{b_{\omega }}^{\gamma }\delta b_{\omega_{k}} \end{array}\right], \end{array}$$
where $\delta b_{a_{k}}$ and $\delta b_{\omega_{k}}$ are obtained from (Equation 7), with $b_{a_{k}}$ and $b_{\omega_{k}}$ discussed in Section 2.3, and $J_{b_{a}}^{\alpha }$ is the submatrix in $F^{\prime }$, whose location corresponds to $\frac{\delta \alpha_{b_{k+1}}^{b_{k}}}{\delta b_{a_{k}}}$. $J_{b_{\omega }}^{\alpha }$, $J_{b_{a}}^{\beta }$, $J_{b_{\omega }}^{\beta }$, and $J_{b_{\omega }}^{\gamma }$ also follow the same notation.

#### 2.1.4. LiDAR Feature Extraction and Distortion Compensation

LiDAR measurements are not made synchronously due to the rotating mechanism inside the LiDAR sensor, and therefore the point cloud $P_{k}$ in the $k^{\mathrm{th}}$ frame suffers from distortion, as shown in Figure 3a. This distortion was compensated for using IMU measurements, as shown in Figure 3b. First, $P_{k}$ is segmented into *N* subframes by azimuthal angle ϕ, where $P_{k}^{i}$ is the $i^{\mathrm{th}}$ subframe for i∈{1,2,…,N}. Second, the transformation matrix from $t_{k}^{i}$ to $t_{k}^{N}$ is defined as $T_{i}^{N}$, and is calculated from the IMU integration as $T_{i}^{N}=T\left(t_{k}^{N}\right)T^{-1}\left(t_{k}^{i}\right)$. Third, by performing subframe-wise transformation, the distortion-compensated point cloud denoted as $P_{k}^{\prime }$ is obtained as
(13)$$P_{k}^{\prime }=\left\{T_{1}^{N}P_{k}^{1},\;T_{2}^{N}P_{k}^{2},\;\ldots ,\;T_{i}^{N}P_{k}^{i}\right\},i=1,\;2,\;...,\;N.$$

The segmentation and $t_{k}^{i}$ are depicted in detail in Figure 4. After performing distortion compensation, the feature points on the planes or the edges in each sweep are extracted using the feature extraction procedure proposed by [10,11].

#### 2.1.5. LiDAR Odometry

In Section 2.1.4, the feature points in each sweep are used to find the corresponding feature points in the last sweep, so that the transformation between each sweep (i.e., $P_{k}^{\prime }$ and $P_{k+1}^{\prime },$ defined in (Equation 13)) can be obtained by minimizing the residual. The procedures are described in detail in [10,11].

### 2.2. Estimator Initialization

In the monocular visual-inertial system [7], the metric scale was recovered through the initialization process. However, the  developed LiDAR-inertial system in this work does not require the initialization process to recover the metric scale thanks to the range measurement from the LiDAR sensor. To help improve the preintegration accuracy, the gyroscope bias $b_{\omega }$ needs to be estimated in Section 2.2.1, and the corrected preintegration can facilitate the estimation of the gravity vector in the first LiDAR frame $g^{l_{0}}$ in Section 2.2.2.

#### 2.2.1. Rotational Alignment

Consider two consecutive frames $b_{k}$ and $b_{k+1}$ in the sliding window, where $l_{0}$ represents the first LiDAR frame. Rotations $q_{b_{k}}^{l_{0}}$ and $q_{b_{k+1}}^{l_{0}}$ are obtained from the given extrinsic parameters $\left(p_{l}^{b},q_{l}^{b}\right)$. Rotations $q_{l_{k}}^{l_{0}}$ and $q_{l_{k+1}}^{l_{0}}$ are from Section 2.1.5. The preintegration ${\bar{\gamma }}_{b_{k+1}}^{b_{k}}$ from (Equation 12) is combined to estimate $\delta b_{\omega }$ by minimizing the following cost function:(14)$$\begin{array}{cc} \min_{\delta b_{\omega }}\sum_{k=0}^{c-1}{∥{\left(q_{b_{k+1}}^{l_{0}}\right)}^{-1}⊗q_{b_{k}}^{l_{0}}⊗{\bar{\gamma }}_{b_{k+1}}^{b_{k}}∥}^{2} & , \end{array}$$
where *c* is the number of frames used for the initialization. Once the gyroscope bias is solved, preintegration terms ${\hat{\alpha }}_{b_{k+1}}^{b_{k}},\;{\hat{\beta }}_{b_{k+1}}^{b_{k}},\;\mathrm{and}\;{\hat{\gamma }}_{b_{k+1}}^{b_{k}}$ will be repropagated using (Equation 12).

#### 2.2.2. Linear Alignment

After computing the gyroscope bias, another important element to consider is the gravity vector. Initialization state $\chi_{I}$ is defined as
(15)$$\chi_{I}=\left[v_{b_{0}},\;v_{b_{1}},\;v_{b_{2}},\;\cdots v_{b_{c-1}},\;g^{l_{0}}\right],$$
which includes the velocities on the body frame of each moment and the gravity vector, where the magnitude of $g^{l_{0}}$ is known.

**Remark** **1.**
*When the UAV is moving during the initialization process, velocity $v_{b_{i}}$ defined in (Equation 15) can be calculated from $p_{b_{i}}^{l_{0}}$, $p_{b_{i+1}}^{l_{0}},$ and $q_{b_{i}}^{l_{0}}.$*


Given two consecutive frames $b_{k}$ and $b_{k+1}$ in the window, $q_{b_{k}}^{l_{0}}$, $q_{b_{k+1}}^{l_{0}}$ and translations $p_{b_{k}}^{l_{0}}$, $p_{b_{k+1}}^{l_{0}}$ obtained are combined with IMU preintegration terms ${\hat{\alpha }}_{b_{k+1}}^{b_{k}}$, ${\hat{\beta }}_{b_{k+1}}^{b_{k}}$ to form the minimization problem
(16)$$\begin{array}{c} \min_{\chi_{I}}\sum_{k=0}^{c-1}{∥{\hat{z}}_{b_{k+1}}^{b_{k}}-H_{b_{k+1}}^{b_{k}}\left[\begin{array}{c} v_{b_{k}}\\ v_{b_{k+1}}\\ g^{l_{0}} \end{array}\right]∥}^{2}, \end{array}$$
to solve state $\chi_{I}$ defined in (Equation 15), where
(17)$$\begin{array}{cc} {\hat{z}}_{b_{k+1}}^{b_{k}} & =\left[\begin{array}{c} {\hat{\alpha }}_{b_{k+1}}^{b_{k}}-R_{l_{0}}^{b_{k}}\left(p_{b_{k+1}}^{l_{0}}-p_{b_{k}}^{l_{0}}\right)\\ {\hat{\beta }}_{b_{k+1}}^{b_{k}} \end{array}\right]\\ \; & =H_{b_{k+1}}^{b_{k}}\chi_{I}+n_{b_{k+1}}^{b_{k}}\\ H_{b_{k+1}}^{b_{k}} & =\left[\begin{array}{ccc} -I\Delta t_{k} & 0 & \frac{1}{2}R_{l_{0}}^{b_{k}}\Delta t_{k}^{2}\\ -I & R_{l_{0}}^{b_{k}}R_{b_{k+1}}^{l_{0}} & R_{l_{0}}^{b_{k}}\Delta t_{k} \end{array}\right], \end{array}$$
and $n_{b_{k+1}}^{b_{k}}$ is the measurement noise. The transformation between each LiDAR measurement, as well as the registered laser scans, will be transformed to the world frame using $g^{l_{0}}$. This is useful when $l_{0}$ frame might not be horizontal, where LiDAR-only odometry may result in a tilted map. After $b_{\omega }$ and $g^{l_{0}}$ are estimated, they can be used as the initial conditions for the tightly coupled LIO in Section 2.3.

### 2.3. Front-End: Tightly Coupled LIO and Mapping

State vector χ, which includes all of the states in the sliding window, is defined as
(18)$$\begin{array}{cc} \chi & =\left[x_{0},\;x_{1},\;x_{2},\;\cdots ,\;x_{m-1},\;x_{l}^{b}\right]\\ x_{k} & =\left[p_{b_{k}}^{w},\;v_{b_{k}}^{w},\;q_{b_{k}}^{w},\;b_{a_{k}},\;b_{\omega_{k}}\right],\;k\in \left\{0,\;1,\;2,\ldots ,\;m-1\right\}\\ x_{l}^{b} & =\left[p_{l}^{b},\;q_{l}^{b}\right], \end{array}$$
where *m* is defined in Table 1, $x_{k}$ is the state at the time when the $k^{th}$ LiDAR measurements are acquired, and $x_{l}^{b}$ are the extrinsic parameters between the LiDAR and IMU.

To estimate the state defined in (Equation 18), the following cost function is optimized to obtain a maximum posteriori estimation:(19)$$\begin{array}{cc} \min_{\chi } & \{{∥r_{p}-H_{p}\chi ∥}^{2}+\sum_{k=0}^{m-1}{∥r_{B}\left({\hat{z}}_{b_{k+1}}^{b_{k}},\;x_{k},\;x_{k+1}\right)∥}_{Q_{b_{k+1}}^{b_{k}}}^{2}\\ \; & +\sum_{j\in L}\rho {∥r_{L}\left({\hat{z}}_{\left(j,f\right)},\;\chi \right)∥}^{2}\}, \end{array}$$
where *L* is a set of indices that characterizes the LiDAR features in the sliding window, which includes two type of sets, namely edge *E* and plane *F*, such that L={E,F}. *f* is the feature correspondence of *j* feature. ρ is a loss function used for outlier rejection, $r_{p}$ and $H_{p}$ are the prior information from marginalization defined in the subsequent analysis, $r_{B}\left({\hat{z}}_{b_{k+1}}^{b_{k}},\;x_{k},\;x_{k+1}\right)$ is the residual for the IMU measurements, and $r_{L}\left({\hat{z}}_{f}^{j},\;\chi \right)$ is the residual for the LiDAR measurements defined in the subsequent analysis. The residuals are described in detail in Section 2.3.1 and Section 2.3.2. To make different types of measurements unitless and scale-invariant, the Mahalanobis norm is applied to (Equation 19). $Q_{b_{k+1}}^{b_{k}}$ is the covariance matrix of the IMU measurement, where $Q_{b_{k+1}}^{b_{k}}$ is obtained by propagating the uncertainty using (Equation 11). The Ceres solver [21] was used to solve the nonlinear problem defined in (Equation 19).

#### 2.3.1. IMU Measurement Model

Replacing true states $\alpha_{b_{k+1}}^{b_{k}},$$\beta_{b_{k+1}}^{b_{k}}$, and $\gamma_{b_{k+1}}^{b_{k}}$ in (Equation 4) with the result from (Equation 12), allows a residual form to be constructed in (Equation 20).
$$r_{B}\left({\hat{z}}_{b_{k+1}}^{b_{k}},\;x_{k},\;x_{k+1}\right)=\left[\begin{array}{c} r_{\alpha_{b_{k+1}}^{b_{k}}}\\ r_{\theta_{b_{k+1}}^{b_{k}}}\\ r_{\beta_{b_{k+1}}^{b_{k}}}\\ r_{b_{a}}\\ r_{b_{\omega }} \end{array}\right]$$
(20)$$=\left[\begin{array}{c} R{\left(q_{b_{k}}^{w}\right)}^{T}\left(p_{b_{k+1}}^{w}-p_{b_{k}}^{w}+\frac{1}{2}g^{w}\Delta t_{k}^{2}-v_{b_{k}}^{w}▵t_{k}\right)-{\bar{\alpha }}_{b_{k+1}}^{b_{k}}\\ 2{\left[{\left({\bar{\gamma }}_{b_{k+1}}^{b_{k}}\right)}^{-1}⊗{\left(q_{b_{k}}^{w}\right)}^{-1}⊗q_{b_{k+1}}^{w}\right]}_{xyz}\\ R{\left(q_{b_{k}}^{w}\right)}^{T}\left(v_{b_{k+1}}^{w}+g^{w}\Delta t_{k}-v_{b_{k}}^{w}\right)-{\bar{\beta }}_{b_{k+1}}^{b_{k}}\\ b_{a_{k+1}}-b_{a_{k}}\\ b_{\omega_{k+1}}-b_{\omega_{k}} \end{array}\right],$$
where ${\left[\cdot \right]}_{xyz}$ denotes the extraction of the imaginary part of the denoted quaternion.

#### 2.3.2. LiDAR Measurement Model

The LiDAR cost function includes frame-to-frame and frame-to-map matching. The frame-to-map matching provides high precision for each state, while the frame-to-frame matching can suppress the variation of the states in the sliding window.

Consider ${\bar{P}}_{j}^{l_{k}}$ to be a feature in the $k^{th}$ LiDAR frame. For frame-to-map matching, ${\bar{P}}_{j}^{l_{k}}$ is represented in the world frame *w* as
(21)$${\bar{P}}_{j}^{w}=T_{b_{k}}^{w}T_{l}^{b}{\bar{P}}_{j}^{l_{k}}.$$

For frame-to-frame matching, point ${\bar{P}}_{j}^{l_{k}}$ is represented in the previous LiDAR frame $l_{k-1}$ as
(22)$${\bar{P}}_{j}^{l_{k-1}}={\left(T_{l}^{b}\right)}^{-1}{\left(T_{b_{k-1}}^{w}\right)}^{-1}T_{b_{k}}^{w}T_{l}^{b}{\bar{P}}_{j}^{l_{k}}.$$

The residuals for edge *E* and plane *F* features are constructed, as shown in Figure 5, and defined as follows.

#### 2.3.3. Residuals for the Edge Features

The point-to-line distance describing the residuals for edge features can be computed as
(23)$$r_{E}\left({\hat{z}}_{\left(j,f\right)}^{t},\;\chi \right)=\frac{\left|\left({\bar{P}}_{j}^{t}-{\bar{P}}_{f_{1}}^{t}\right)\times \left({\bar{P}}_{j}^{t}-{\bar{P}}_{f_{2}}^{t}\right)\right|}{\left|{\bar{P}}_{f_{1}}^{t}-{\bar{P}}_{f_{2}}^{t}\right|},$$
where ${\bar{P}}_{j}^{t}$ is ${\bar{P}}_{j}^{l_{k}}$ represented in *t* frame, *t* can represent either *w* or $l_{k-1}$ using (Equation 21) or (Equation 22), respectively. ${\bar{P}}_{f_{1}}^{t}$ is the closest edge feature to ${\bar{P}}_{j}^{t}$, and ${\bar{P}}_{f_{2}}^{t}$ is the second closest point.

#### 2.3.4. Residuals for the Plane Features

The point-to-plane distance known as the Hesse normal form can be computed
(24)$$r_{F}\left({\hat{z}}_{\left(j,f\right)}^{t},\;\chi \right)=n_{f}^{t}\cdot {\bar{P}}_{j}^{t}+d_{f}^{t},$$
where $n_{f}^{t}$ is the normal vector of the closest plane to ${\bar{P}}_{j}^{t}$, and $d_{f}^{t}$ is the distance from the closest plane to the origin of frame *t*.

The residuals described in (Equation 23) and (Equation 24) are applied to construct one of the residuals defined in (Equation 19) as
(25)$$\begin{array}{cc} \sum_{j\in L}\rho {∥r_{L}\left({\hat{z}}_{\left(j,f\right)},\;\chi \right)∥}^{2} & =\sum_{j\in E}\rho {∥r_{E}\left({\hat{z}}_{\left(j,f\right)}^{w},\;\chi \right)∥}^{2}\\ \; & +\sum_{j\in F}\rho {∥r_{F}\left({\hat{z}}_{\left(j,f\right)}^{w},\;\chi \right)∥}^{2}\\ \; & +\sum_{j\in F}\rho {∥r_{F}\left({\hat{z}}_{\left(j,f\right)}^{l_{k-1}},\;\chi \right)∥}^{2}, \end{array}$$
where $r_{E}\left({\hat{z}}_{\left(j,f\right)}^{w},\;\chi \right)$ is the residual of the edge feature for frame-to-map matching, $r_{F}\left({\hat{z}}_{\left(j,f\right)}^{w},\;\chi \right)$ is the residual of the plane feature for frame-to-map matching, and  $r_{F}\left({\hat{z}}_{\left(j,f\right)}^{l_{k-1}},\;\chi \right)$ is the residual of the plane feature for frame-to-frame matching. The residual of the edge feature for frame-to-frame is not considered, since it does not help for boosting the accuracy of RTLIO.

#### 2.3.5. Marginalization

In order to reduce the computational complexity and preserve the history information, a marginalization procedure needs to be applied to the sliding window method. This marginalization aims to keep the most-recent frame in the window, and the Schur complement is applied to construct a prior term based on marginalized measurements. The detail can be referred to [22,23]. A factor graph of such a system is shown in Figure 6. The frame-to-map constraints do not influence the adjacent states, and so only the frame-to-frame constraints are considered.

Combining all of the residuals and solving the cost function defined in (Equation 19) yields the best estimation of states. The local map is then obtained based on the current state estimation, by applying an appropriate algorithm [11].

### 2.4. Back-End: Loop Closure and Pose-Graph Optimization

The optimization-based approach provides sufficient accuracy in an indoor environment, but for large-scale cases, it is inevitable that accumulated drift will occur due to various factors, such as extrinsic parameters between the LiDAR and IMU, the asynchronous sampling of measurements, and inaccurate data association during LiDAR matching. One way to correct for such drift is using loop closure. This method starts with identifying the previously visited places. Once a loop is detected by computing feature correspondences, a relocalization process tightly integrates these constraints into a cost function. This procedure minimizes drift and achieves much smoother state estimation. After  the loop closure and relocalization are performed, the sliding window shifts and aligns with the past poses. Then, a pose-graph optimization algorithm can match all keyframes, in order to minimize the drift and ensure the global consistency of the system. These processes might not influence the current state estimation, but the optimized pose-graph can facilitate the consistency of global map reconstruction after performing the state estimation.

#### 2.4.1. Loop Closure

The loop closure algorithm is described in Algorithm 1. Once a frame is marginalized from the sliding window, its point cloud in the body frame, $P_{m}$, and its pose, $T_{m}$, will be fed into the loop closure algorithm. If the L2-norm between $T_{m}$ and the pose at the lastest keyframe is higher than an Euclidean distance threshold, the marginalized frame is considered as a new keyframe. In this way, the keyframes are kept uniformly distributed in the space. Then, KD-tree search with search radius of *r* is performed if the keyframe database, $\left(D_{T},D_{P}\right)$, which is the set of the keyframe pose and point cloud, is not empty. $T_{m}^{{}^{\prime }}$ from $D_{T}$ is the closest keyframe transformation matrix to $T_{m}$. If $T_{m}^{{}^{\prime }}$ can be found in $D_{T}$, the loop is assumed to be detected. Then, $T_{ICP}$ is obtained by matching $P_{m}$ with the local map, M, based on the threshold of the point-to-point RMSEs. *M* is the local map constructed by registering keyframe point cloud $D_{P}$ to the world frame based on $D_{T}$.
**Algorithm 1:** Loop closure algorithm.**Input:** $T_{m},P_{m}$ from the sliding window**Output:** $T_{ICP}$1:**if** $(T_{m},P_{m}).$isKeyframe() **then**2: **if** $D_{T}\neq \phi$ or $D_{P}\neq \phi$ **then**3:   $T_{m}^{{}^{\prime }}\leftarrow$ KDtree.RadiusSearch$(T_{m},D_{T},r);$4:   M← registerPointCloud$(D_{T},D_{P});$5:   **if** $T_{m}^{{}^{\prime }}\neq 0$ **then**6:    $T_{ICP}\leftarrow$ ComputeICP$(P_{m},M,T_{m});$7:   **end if**8:  **end if**9:  $T_{m}^{{}^{\prime }}\leftarrow 0$10:  $D_{T}=D_{T}\cup T_{m}$11:  $D_{P}=D_{P}\cup P_{m}$12:**end if**

#### 2.4.2. Tightly Coupled Relocalization

As long as the current pose in the world coordinate is obtained, $T_{ICP}$ and *M* are fed back to the RTLIO module for state correction. The relocalization scheme is modified from (Equation 19) by solving the following cost function:$$\begin{array}{cc} \; & \min_{\chi }\{{∥r_{p}-H_{p}\chi ∥}^{2}+\sum_{k=0}^{m-1}{∥r_{B}\left({\hat{z}}_{b_{k+1}}^{b_{k}},\;x_{k},\;x_{k+1}\right)∥}_{Q_{b_{k+1}}^{b_{k}}}^{2}\\ \; & +\sum_{j\in L}\rho {∥r_{L}\left({\hat{z}}_{\left(j,f\right)},\;\chi \right)∥}^{2}+\sum_{j\in L}\rho {∥r_{M}\left({\hat{z}}_{\left(j,m\right)},\;\chi \right)∥}^{2}\} \end{array}$$
with the constraint
(26)$$\begin{array}{cc} r_{M}\left({\hat{z}}_{\left(j,m\right)},\;\chi \right) & ={\bar{P}}_{j}^{w}-{\bar{P}}_{m}^{w}\\ \; & =R_{b_{k}}^{w}\left(R_{l}^{b}{\bar{P}}_{j}^{l_{k}}+p_{l}^{b}\right)+p_{b_{k}}^{w}-{\bar{P}}_{m}^{w}, \end{array}$$
where ${\bar{P}}_{m}^{w}\in M$ is the closest point to ${\bar{P}}_{j}^{w}$ in the global map. By solving the modified cost function, the current states can be used for relocalization in the global map.

#### 2.4.3. Global Pose-Graph Optimization

Due to the LiDAR inertial setup, roll and pitch angles are fully observable once gravity and the bias are estimated. Therefore, the accumulated drift only occurs in the other four degrees of freedom (*x*, *y*, *z*, yaw) and can be reduced by solving keyframe states in the pose-graph. Every keyframe state serves as a vertex in the pose-graph, and two types of edges between the vertices are utilized. The pose-graph is illustrated in Figure 7.

#### 2.4.4. Sequential Edge

A sequential edge represents the relative transformation between each keyframe, which is obtained from the RTLIO results. Considering a keyframe *j* and its previous keyframe *i*, the sequential edge is defined as $s_{j}^{i}=\left\{{\hat{p}}_{j}^{i},\;{\hat{\psi }}_{j}^{i}\right\}$, where ${\hat{p}}_{j}^{i}$ and ${\hat{\psi }}_{j}^{i}$ denote the relative position and the relative yaw angle, respectively:(27)$$\begin{array}{cc} {\hat{p}}_{j}^{i} & ={\hat{R}}_{i}^{w}\left({\hat{p}}_{j}^{w}-{\hat{p}}_{i}^{w}\right)\\ {\hat{\psi }}_{j}^{i} & ={\hat{\psi }}_{j}^{w}-{\hat{\psi }}_{i}^{w}. \end{array}$$

If the current keyframe *j* has a corresponding keyframe *i*, the loop closure edge is defined as $h_{j}^{i}=\left\{{\hat{p}}_{j}^{i},\;{\hat{\psi }}_{j}^{i}\right\}$ which is obtained in Section 2.4.1. Then loop closure edge will be added to the pose-graph as an additional constraint.

#### 2.4.5. Pose-Graph Optimization for Four Degrees of Freedom

State vector $\chi_{p}$ of the pose-graph is defined as
(28)$$\begin{array}{cc} \chi_{p} & =\left[x_{0},\;x_{1},\;\cdots ,\;x_{n-1}\right]\\ x_{k} & =\left[p_{k}^{w},\;\psi_{k}^{w}\right],\;k\in \left\{1,\;2,\;\ldots ,\;n-1\right\}, \end{array}$$
where *n* is the number of vertices in the pose-graph.

To find state $\chi_{p}$ defined in (Equation 28), the cost function is formulated as
(29)$$\min_{\chi_{p}}\left\{\sum_{s_{j}^{i}\in S}{∥r_{ij}∥}^{2}+\sum_{h_{j}^{i}\in H}\rho {∥r_{ij}∥}^{2}\right\},$$
where the residuals of the sequential edge and the loop closure edge between keyframes *i* and *j* are defined as
(30)$$\begin{array}{cc} r_{ij}\left(p_{i}^{w},\;\psi_{i}^{w},\;p_{j}^{w},\;\psi_{j}^{w}\right) & =\\ \; & \left[\begin{array}{c} R{\left({\hat{\phi }}_{i}^{w},\;{\hat{\theta }}_{i}^{w},\;\psi_{i}^{w}\right)}^{-1}\left(p_{j}^{w}-p_{i}^{w}\right)-{\hat{p}}_{j}^{i}\\ \left(\psi_{j}^{w}-\psi_{i}^{w}\right)-{\hat{\psi }}_{j}^{i} \end{array}\right], \end{array}$$
where ${\hat{\phi }}_{i}^{w}$ and ${\hat{\theta }}_{i}^{w}$ are the roll and pitch angles, respectively, converted from $q_{b_{k}}^{w}$ defined in (Equation 18). The loss function ρ is applied to penalize wrong connections of the loop closure edges. Once pose-graph optimization is completed, all keyframe states are updated. The global map is then updated by registering the keyframe point cloud according to the states.

## 3. Experiment Results and Discussions

A series of experiments (https://github.com/ChadLin9596/ncrl_lio, accessed on 2 June 2021) were conducted to analyze the performance of the developed RTLIO algorithm and compare it with the current state-of-the-art algorithms. To demonstrate the real-time capability of our system, multiple indoor tests are presented in Section 3.1. The KITTI dataset was used to compare the results with real-world benchmarks; the results are discussed in Section 3.2.

### 3.1. Indoor Flight Test

During the experiments, multiple threads were utilized to achieve the desired performance in real time. The first thread performed distortion compensation and feature extraction from LiDAR measurements, as described in Section 2.1.4. The second thread took those features and computed the incremental motion, as described in Section 2.1.5. The third thread (described in Section 2.3) executed the RTLIO algorithm that solves the states based on the initial guess from the second thread. The RTLIO generated two types of odometry defined in Section 1.3: (i) LiDAR-rate pose, and (ii) the IMU-rate pose, which can be obtained with minimal delay. This means that the high-frequency pose can be directly used for real-time feedback control.

The precision and computation time are discussed and compared with other LiDAR-based methods in Section 3.1.2, for experiments conducted in the laboratory with the OptiTrack motion capture system as the ground truth. The flight tests with RTLIO and the other methods are presented in Section 3.1.3.

#### 3.1.1. System Setup

The quadcopter setup used in this work is shown in Figure 8. It comprised a 16-beam LiDAR system (Velodyne VLP-16, 10 Hz), IMU (400 Hz), and Intel NUC (NUC8i7BEH) with an i7-8559U CPU running at 2.70 GHz and 20 GB of memory. The RTLIO algorithm is implemented on the board to perform state estimation in real time.

#### 3.1.2. Precision and Time Cost

Data recorded with quadcopter flying in circular trajectories in the laboratory were used as the input to LOAM (cf. [11]), ALOAM (cf. [11] (https://github.com/HKUST-Aerial-Robotics/A-LOAM, accessed on 1 April 2021), and RTLIO to conduct postprocessing. The comparison of performance is shown in Figure 9, where ALOAM is an extension of LOAM produced by HKUST. The relative pose errors (RPEs) and the time costs of the methods are listed in Table 2, which indicates that the odometry from RTLIO had lower RPEs, but a greater time cost. The methods discussed in this section are able to estimate the pose state to a certain precision, and the impact of the time costs is discussed in Section 3.1.3. The IMU-rate poses from RTLIO cannot be compared with poses from ALOAM and LOAM, because ALOAM and LOAM cannot generate high frequency poses.

#### 3.1.3. Indoor Flights

Figure 10 shows the results of RTLIO along the *x*, *y*, and *z* axes, from take off to landing. These results show that high-performance localization is crucial to the real-time feedback control of the quadrotor and trajectory tracking, and the time delays from RTLIO and LOAM are compared in Figure 11. The computation time for RTLIO is 0.2944 ms, and the delay is small enough for feedback control.

#### 3.1.4. Indoor Flight with an Obstacle

Indoor flight experiments were also conducted with the quadcopter flying along a corridor with the localization obtained by RTLIO. Figure 12a shows the setup of the indoor test environment, Figure 12b shows the top view of the map in the xy plane and the trajectory of the UAV, starting from the “WORLD” to “IMU”. These tests and the results presented in Section 3.1.3 demonstrate the capability of the RTLIO algorithm to perform localization for the feedback control of the quadcopter and trajectory tracking, either in a laboratory or corridor environment, and that the generated mapping is reliable.

### 3.2. KITTI Dataset Evaluation

The developed RTLIO was also evaluated using KITTI dataset, which includes measurements from an inertial navigation system (OXTS RT3003), which provides the ground-truth pose and IMU measurements at 100 Hz, a 64-beam LiDAR (Velodyne HDL-64E, 10 Hz), two grayscale (Point Grey Flea 2 FL2-14S3M-C, 10 Hz), and two color cameras (Point Grey Flea 2 FL2-14S3C-C, 10 Hz). In this test, only the IMU and the LiDAR were used to evaluate our algorithm.

#### 3.2.1. Front-End Performance

The RTLIO in the front-end did not include loop closure and pose-graph optimization. The results show that the average errors of the translation and rotation along the given path were 1.8560 % and 0.0043 deg/m, respectively, as reported by the KITTI evaluation. The average translation and rotation errors over different lengths in each sequence are shown in Figure 13.

#### 3.2.2. Full Closed-Loop Performance

After adding the back-end to the RTLIO, the full pipeline was also evaluated using the KITTI dataset. The overall results show that the average errors of the translation and rotation along the given path are 1.6392% and 0.0035 deg/m, respectively. Figure 14 shows that the RTLIO including the back-end effectively reduces the errors compared with the front-end in Figure 13.

The APE (Absolute Pose Error) evaluated with EVO (https://github.com/MichaelGrupp/evo, accessed on 1 April 2021) is listed in Table 3, in which the sequences that contain a trajectory without loop closure are marked with an *.

The results in Table 3 indicate that the RTLIO with back-end can outperform RTLIO in APE (especially in the sequence with loop closure).

### 3.3. Time Consumption

The time consumption of each module in indoor flight test and KITTI datasets using an Intel i7-7700 CPU with 24~GB of memory is listed in Table 4. Threads 1–3 are used for computing the front-end of the RTLIO, and thread 4 is used for the back-end, which also reconstructs a globally consistent map. However, the RTLIO was unable to run in real time using the KITTI datasets, since scan matching is more difficult in the outdoor environment. Additionally, the time consumption increases with the increase of the number of the channels of the LiDAR. The higher the number of channels, the higher the resolution (i.e., 16 channels for VLP-16, 32 channels for HDL-32E), according to the website from Velodyne: (https://velodynelidar.com/products/hdl-32e, accessed on 2 June 2021).

## 4. Conclusions and Future Work

The RTLIO developed in this work can generate accurate and reliable odometry information in real time, and the initialization process is performed when the UAV is already in motion. The developed RTLIO method uses LiDAR and IMU to generate high-frequency odometry with improved performance compared to the methods that only use LiDARs. Moreover, a consistent and accurate global map is constructed using the loop closure and pose-graph optimization method. Experiments were conducted with the quadrotor in indoor environment and using KITTI dataset, and the results demonstrated that the RTLIO can outperform ALOAM and LOAM in terms of exhibiting a smaller time delay and greater flight stability. The RTLIO with back-end algorithm can outperform the RTLIO with only front-end algorithm, since the accumulated drift can be reduced by the developed pose graph.

Future works include designing a more stable initialization method to deal with diverse situations. In addition, detection algorithms can be integrated into the method for removing feature points on moving objects. Finally, integrating vision sensors with the current system to improve the precision of odometry will be conducted to increase the stability of pose estimation along the *z* axis.

## Figures and Tables

**Figure 1 sensors-21-03955-f001:**
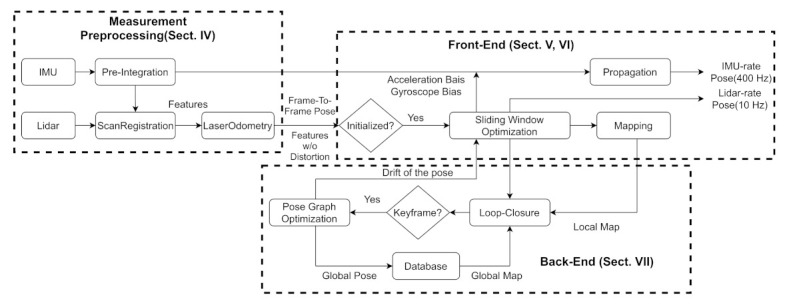
Architecture of the developed RTLIO.

**Figure 2 sensors-21-03955-f002:**
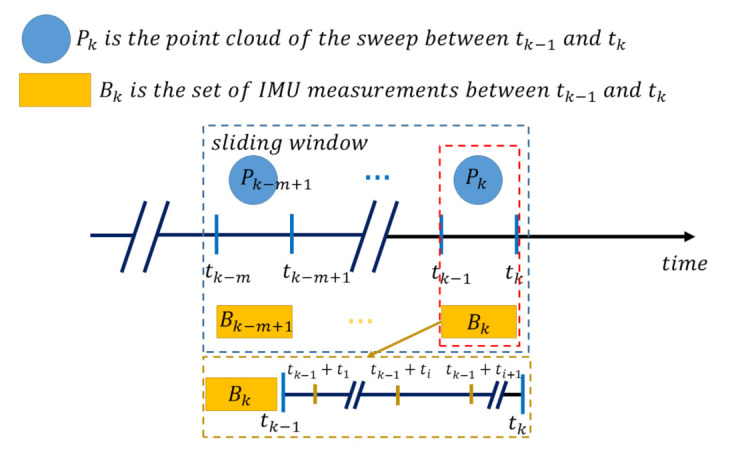
Time alignment between LiDAR point cloud $P_{k}$ and the set of the IMU measurements $B_{k}$.

**Figure 3 sensors-21-03955-f003:**
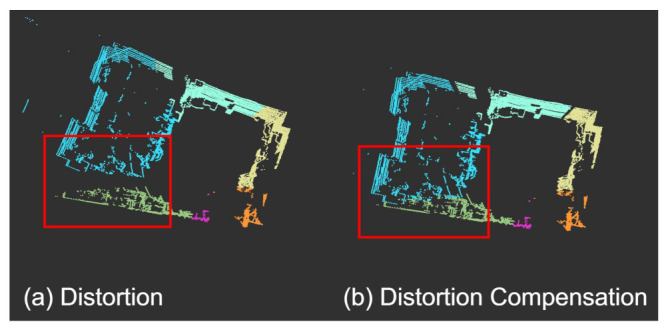
Calibrated results: (**a**) the point cloud suffered from distortion when LiDAR is moving, and (**b**) the result obtained after distortion compensation. Different colors indicate subframes in one sweep.

**Figure 4 sensors-21-03955-f004:**
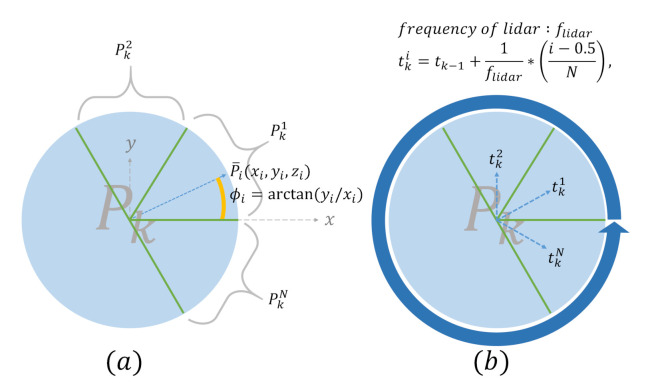
(**a**) Each subframe of $P_{k}$ is defined as $P_{k}^{i}.$ (**b**) The time of the $i^{\mathrm{th}}$ subframe is defined as $t_{k}^{i}$ for i∈{1,...,N}.

**Figure 5 sensors-21-03955-f005:**
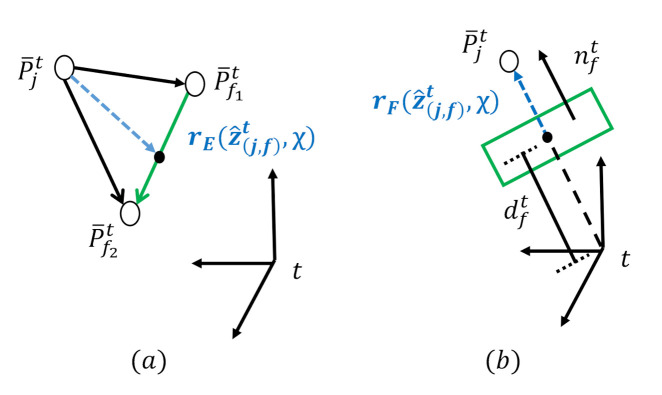
Illustration of residuals for edge and plane features. Residuals are shown by the blue lines. (**a**) The residual of edge feature ${\bar{P}}_{j}^{t}$ with the corresponding line shown in green that is formed by ${\bar{P}}_{f_{1}}^{t}$ and ${\bar{P}}_{f_{2}}^{t}$. (**b**) The residual of the plane feature ${\bar{P}}_{j}^{t}$ with the plane shown in green that is formed by the feature correspondences.

**Figure 6 sensors-21-03955-f006:**
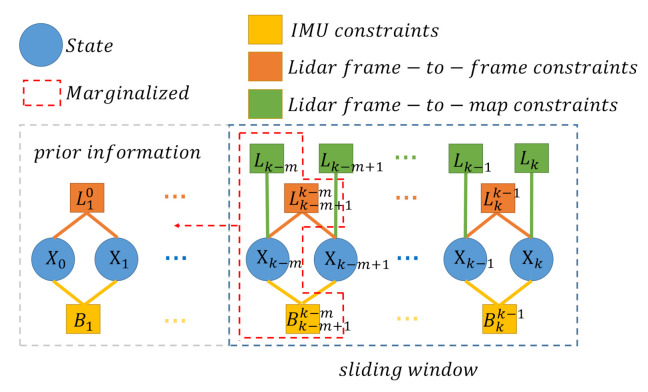
Illustration of the factor graph and the marginalization strategy. The oldest frame in the sliding window will be marginalized into prior information after optimizing (Equation 19).

**Figure 7 sensors-21-03955-f007:**
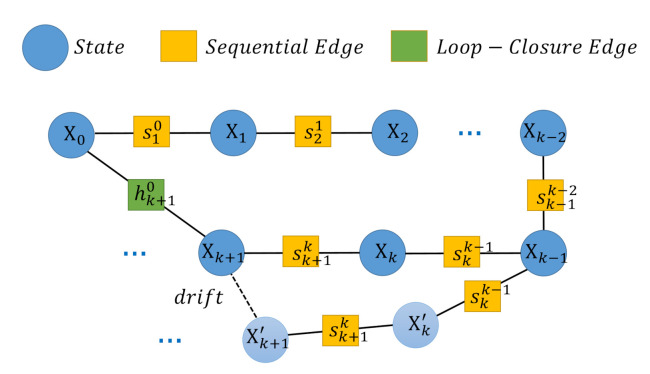
Constructing a pose-graph: every node in the graph represents the state of a keyframe. *S* is the set of the sequential edges, where $S=\left\{s_{1}^{0},\;s_{2}^{1},\;\cdots ,\;s_{k+1}^{k},\;\cdots \right\}$. *H* is the set of loop closure edges, where $H=\left\{h_{k+1}^{0},\;\cdots \right\}$.

**Figure 8 sensors-21-03955-f008:**
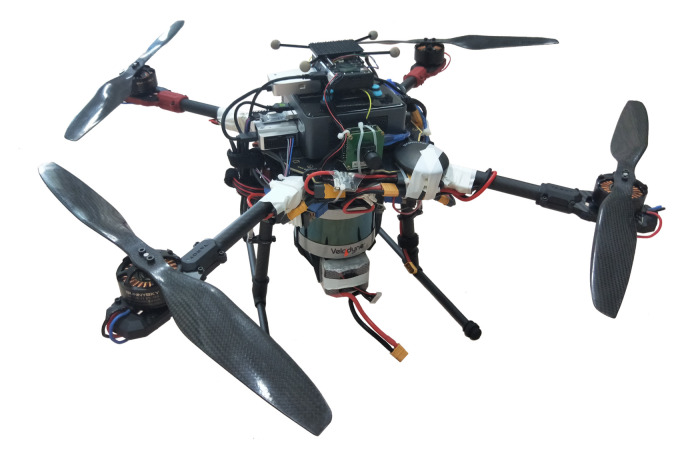
The quadcopter used for the indoor flight tests.

**Figure 9 sensors-21-03955-f009:**
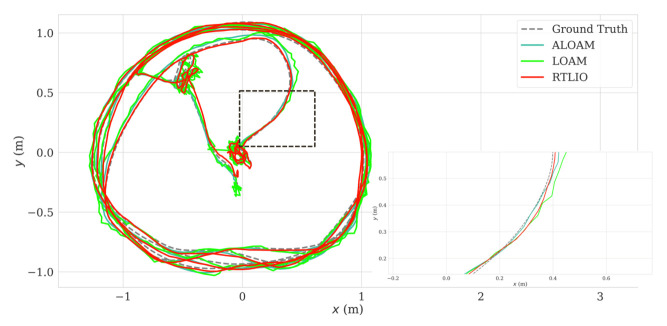
The comparison of trajectory with LOAM, ALOAM, RTLIO.

**Figure 10 sensors-21-03955-f010:**
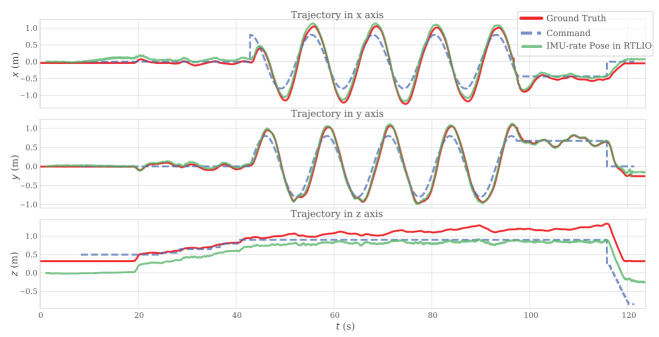
Flight trajectory with RTLIO along the *x*, *y*, and *z* axes.

**Figure 11 sensors-21-03955-f011:**
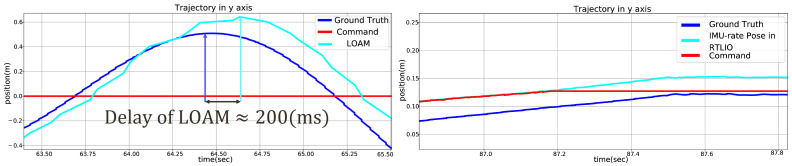
Time delays for (**left**) LOAM, (**right**) IMU-rate pose in RTLIO.

**Figure 12 sensors-21-03955-f012:**
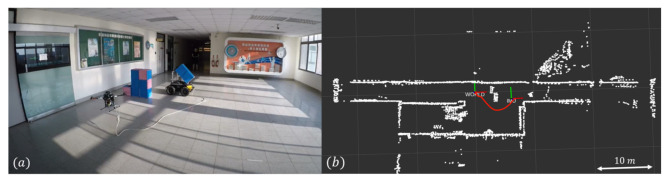
Results for indoor flying with a corridor: (**a**) setup (**b**) top view of the map and the trajectory of the entire flight (from WORLD to IMU).

**Figure 13 sensors-21-03955-f013:**
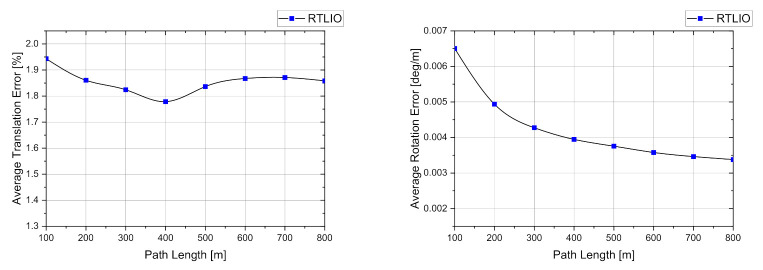
Average translation and rotation errors of the front-end evaluated over different lengths in the KITTI dataset for sequence from 00 to 10.

**Figure 14 sensors-21-03955-f014:**
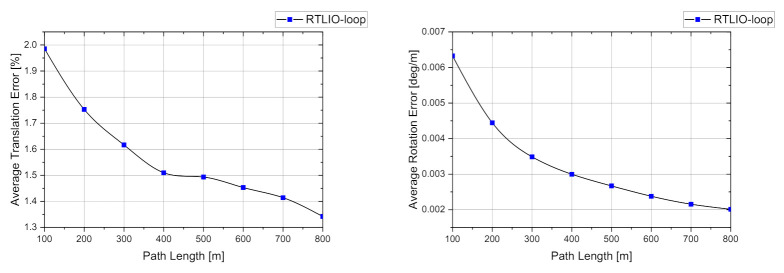
Average translation and rotation errors of the full pipeline evaluated over different lengths in the KITTI dataset for sequences 00 to 10.

**Table 1 sensors-21-03955-t001:** Notation.

Index	Note
$p\in R^{3}$	position
$v\in R^{3}$	velocity
*q*	quaternion
$\theta \in R^{3}$	Euler angle
R∈SO(3)	rotation matrix
$T\in R^{44}$	transformation matrix
$\omega \in R^{3}$	angular velocity
$a\in R^{3}$	linear acceleration
$g\in R^{3}$	gravity
$b_{a},b_{\omega }\in R^{3}$	acceleration and gyroscope bias
$n_{a},n_{\omega }\in R^{3}$	acceleration and gyroscope noise
*P*	point cloud
$\bar{P}\in R^{3}$	a point in *P*
*b*	body frame
*w*	world frame
*l*	LiDAR frame
${\left(\cdot \right)}^{i}$	state representation in ith frame
${\left(\cdot \right)}_{t}$	state at time *t*
$\hat{\left(\cdot \right)}$	nominal state
|·|	cardinality of the denoted argument
m∈R	number of frames in sliding window

**Table 2 sensors-21-03955-t002:** Comparison of RMSE of RPE and average time costs for the RTLIO, ALOAM and LOAM methods.

Method	Number of Frames	Translation (m)	Rotation (deg)	Computation Time (ms)
LOAM (10 Hz)	1203	0.0599	1.4218	67.5977
ALOAM (10 Hz)	1224	0.0078	0.3955	61.1810
RTLIO (10 Hz)	1224	0.0066	0.1881	96.3577

**Table 3 sensors-21-03955-t003:** Translation and rotation of APE in the KITTI dataset.

	RTLIO		RTLIO with Back-End	
**Sequence**	**Translation (m)**	**Rotation (deg)**	**Translation (m)**	**Rotation (deg)**
00	9.4542	2.5884	**1.8196**	**0.7324**
* 01	**27.5966**	**8.0052**	31.3346	9.2077
02	10.3673	1.5718	**5.8435**	**1.3680**
* 04	1.8050	**1.2320**	**1.0295**	1.3538
05	3.5576	1.7812	**0.9164**	**0.4610**
06	5.7340	2.9552	**1.4797**	**0.6996**
07	1.2983	0.7238	**0.9850**	**0.6497**
08	22.4302	4.0389	**10.2060**	**2.1994**
09	20.9436	5.8630	**3.4717**	2.3290
* 10	2.3719	1.2684	**2.3041**	**1.2050**

**Table 4 sensors-21-03955-t004:** Time Statistics.

Thread	Module	Time (ms)		Rate (Hz)
		Indoor	KITTI	
1	feature extraction	6	25	10
2	frame-to-frame odometry	15	65	10
3	sliding window optimization	65	350	10
4	loop closure	130	200	X
	pose-graph optimization	10	120	X

## Data Availability

Data available in a publicly accessible repository that does not issue DOIs.

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
