# Peer review of "RTLIO: Real-Time LiDAR-Inertial Odometry and Mapping for UAVs"

_sensors, 2021, doi:10.3390/s21123955_

Round 1

Reviewer 1 Report

Authors presented an interesting approach for inertial odometry and mapping for UAVs.  Overall, the presentation was detailed in the methodology development and the experimental design while still being easy to follow.  I few notes for improvement of the paper are:

line 22 - ref 4 is assumed to be "the" reference for sensor fusion, which it is not, please add more foundational/background refs.

Figure 1 - it is not reccommended to outline anything in figure in red unless it is to attract the reader's eye to a specific and critical component.  Change figure outline color.

line 53 - please explain why KITTI dataset is chosen (instead of some other dataset) and add a ref on where the reader can find more info on the dataset itself

Line 61 - the Related Works section is normally part of the introduction, it's not needed to be a seperate section

Line 88 - Fig 1 is referenced and discussed here, it's easier on readers for the figure to be on the same page as the discussion, please reorganize if possible.

Equations 1, 2, and 6 - please define the two box symbols, and the script N

And lastly, there are typos in lines 281 and 290.

Reviewer 2 Report

This research developed a real-time LiDAR inertial odometer system (RTLIO) to achieve high-precision and high-frequency odometry and mapping for feedback control of UAVs. When compared to the traditional LIO approach, the initialization process can be conducted even when the device is stationary. Research results revealed that the proposed RTLIP method outperforms the LIP method with a smaller drift. The reviewer believes that the current version of the manuscript is not yet ready for publication; the authors are encouraged to consider the following comments and suggestion and revise the manuscript accordingly.

  1. The authors should streamline the Abstract section. Currently, it is very short and does not cover all of the required information. The Abstract section should focus on explaining why the research is needed, what the research is about, what the methodology is, and what the conclusion is. Do not include any unnecessary information but the required information must be provided.
  2. The authors should consider reorganizing the manuscript to include the following sections: Introduction, Background, Methodology, Results and Discussion, and Conclusions. The Introduction section should focus on introducing the research objectives and stating the research questions that need to be answered, while the Background section should focus on reviewing of related literature and presenting the process of finding the research gap. The contents in the Overview section should be moved to the Introduction section. The authors should also expand their literature review. For example, For example, the authors should read and cite the research of “The impact of small unmanned airborne platforms on passive optical remote sensing: a conceptual perspective” to discuss the trend of UAVs in scientific research.
  3. The authors should have a researcher that has a remote sensing background to proofread the manuscript. Many terms used in the manuscript are remote sensing related and the authors did not use them correctly. For example, on page 18 line 284, the authors stated that “Additionally, the time consumption increases with the increase of the resolution of lidar measurements”. What resolution do you refer to? There are four resolutions for aerial imagery, including spatial resolution, spectral resolution, temporal resolution, and radiometric resolution. The authors should use spatial resolution instead of resolution if that is what they want to indicate. In addition, LiDAR is light detection and ranging, and normally it is written as LiDAR, not lidar. Additionally, LiDAR is usually discussed in terms of vertical accuracy and point density, not spatial resolution.
  4. The authors should include a documentation for explaining their algorithms. Such a documentation will assist researchers in replicating the proposed method. The authors also need to go through the equations to make sure all elements in the equations are denoted.
  5. The manuscript needs to be reformatted since it does not meet the formatting requirements of the journal.
  6. All figures and tables need to be improved. Currently, they do meet the requirements of the journal and they are in poor quality. Please refer to the published articles for figure and table improvement. In addition, the figures are not readable and legible. If at all possible, the authors should use vector image for figure presentation.

Reviewer 3 Report

This is a well-written and very interesting paper. It is a bit above my area of expertise but I do believe it should be published since it seems to be a good starting point for work on the described method to make it more universal

Round 2

Reviewer 2 Report

The authors have addressed all my comments.